# Coloration Modeling and Processing of Commodity Plastic Buttons in Supercritical Carbon Dioxide

**DOI:** 10.3390/ma16030907

**Published:** 2023-01-17

**Authors:** Kota Kobayashi, Tierong Bai, Kazuhiro Tamura, Kaoru Tada, Jun Yan, Laijiu Zheng

**Affiliations:** 1School of Natural System, College of Science and Engineering, Kanazawa University, Kakuma-machi, Kanazawa 920-1192, Japan; 2School of Textile and Material Engineering, Dalian Polytechnic University, Dalian 116034, China

**Keywords:** scCO_2_, dyeing, plastic button, solubility, diffusion–adsorption model, *K*/*S* value

## Abstract

We examined the color processing of the commodity plastic buttons made of acrylic, polyester, nylon, and casein with scCO_2_. The buttons’ dyeing color depth (*K*/*S*) was measured over a wide range of scCO_2_ and correlated accurately with the response surface method. Moreover, we measured the solubility of C.I. Disperse Red 22 in scCO_2_ to formulate a dye-sorption model for the *K*/*S* value in the color processing of the plastic buttons. Finally, the dye-sorption model for the *K*/*S* value combining the dye solubility in scCO_2_ with the dye diffusion inside the buttons successfully represented the color processing of the buttons.

## 1. Introduction

Supercritical carbon dioxide (scCO_2_) is applied for several industrial processes because it is nontoxic, non-flammable, harmless, low-price, and much more. It is often used for extracting functional antioxidant components from natural plants in food industries and separating side products oligomers and solvent residues in pharmaceutical and chemical companies [1,2]. After the first development by the Schollmeyer group, scCO_2_ dyeing has been paid attention to in the textile industry as an environmentally friendly amd waterless coloration technology [3,4,5,6,7]. The scCO_2_ dyeing process has advantages over conventional water dyeing from a green and energy-saving viewpoint. At supercritical conditions, scCO_2_ has both a high density like liquid, and a high diffusivity and low viscosity like gas. It has the ability to dissolve dyestuffs into scCO_2_ due to high density like water, and its diffusivity is 10 to 100 times higher than water. Therefore, scCO_2_ makes the polymer swollen, and dyes can diffuse into the polymer. Since swollen polymer networks in scCO_2_ cause the glass transition temperature depression of polymers, dyes diffuse more easily over the glass transition temperature, and it puts into practice the high dyeing concentration in a short time. Furthermore, we can reduce much more energy and costs by scCO_2_ dyeing because of the absence of a wastewater treatment and a drying process. According to the world’s first supplier of industrial CO_2_ dyeing equipment company, DyeCoo Textile Systems [8], they could cut down about 43% of the energy used in the dyeing treatment for polyester fabrics, comparing water dyeing with scCO_2_ dyeing. Unfixed dye and more than 90% of CO_2_ can be recycled and reused.

scCO_2_ dyeing has been applied to several resins and fibers, synthetic or natural, so far. For example, dyeing polyester (PET) fibers were investigated intensively. It elucidated that the PET could reach the saturation of dyeing at 80 °C, 3500 psi for 30 min, and its dyeing result is similar to those obtained in an aqueous solvent at 120 °C for 1 h [9]. Moreover, it revealed the difference in the dyeing concentration and condition that reach the (equilibrium due to the type of dyestuff, and it calculated the partition coefficient for the dyes between the fluids and the polymer phase from the new solubility data. The dye uptake of the fabrics impregnated by scCO_2_ has been described in terms of the dyeing temperature, pressure, time, and dye concentration in scCO_2_. Moreover, it has extended to polycarbonate (PC) [10], meta-aramid [11], cotton fabrics [12,13], and protein fibers [14]. Industrial products such as zipper tapes [15] and fabric ropes [16] made by the scCO_2_ dyeing process have also been developed.

Previously, we carried out an optimization study on scCO_2_ dyeing for synthetic and natural resin buttons using only name experiment runs based on the experimental design method of Taguchi [17]. In the present work, we aim to examine the color processing of plastic buttons made of acrylic, polyester, and casein in scCO_2_ to control the degree of dyeing for the plastic buttons. Therefore, we carried out the measurements on scCO_2_ dyeing for synthetic and natural resin buttons in order to add an extension to the number and ranges of experimental points measured in the previous results. Moreover, we measured the solubility of C.I. Disperse Red 22 in scCO_2_, which is a fundamental property of the mass transfer of the dye into the plastic buttons and is necessary in the correlation of the degree of dyeing for the plastic buttons. From the above, we developed a response surface model to represent the *K*/*S* values for plastic buttons in terms of the solubility of dyestuff y, pressure *P*, temperature *T*, and treatment time *t*, and clarified the effects of the dyeing color processing on the plastic buttons using the regression model. Furthermore, we formulated a dye-sorption model to examine the *K*/*S* values for plastic buttons by combining the dye solubility in scCO_2_, the adsorption equilibrium on the button surface, and the diffusion into the buttons with the amount of dye sorption in the button, to understand the coloration processing of plastic buttons in scCO_2_.

## 2. Experimental

### 2.1. Chemicals

C.I. Disperse Red 22 (C_20_H_13_NO_2,_ 1-phenylaminoanthraquinone) with a purity of 99.7 wt % was supplied by Kiwa Chemical (Wakayama, Japan). The melting point of the dyestuff measured by a melting-point apparatus (MP-21, Yamato Scientific Co. Ltd., Tokyo, Japan) was 139.5 ℃ CO_2_ with a purity of 99.9 wt %, which was supplied by Uno Sanso Co (Ishikawa, Japan). Ethanol with a purity of 99.9 wt %, which was provided by Japan Alcohol Corp (Tokyo, Japan). Commercial synthetic resin buttons made of acrylic, polyester, casein, and nylon were purchased from Shigejin Co. Ltd (Kyoto, Japan). The average molecular mass and polydispersity index and glass transition temperature of the buttons was reported earlier [17]. The glass transition temperatures T_g_ for the acrylic, polyester, casein, and nylon polymers were estimated to be 96.2–101.4 °C, 53.0–56.9 °C, 60.5 °C, and 64.4 °C, respectively.

### 2.2. Solubility Data

The solubility data of C.I. Disperse Red 22 in scCO_2_ were measured by a flow-type apparatus shown in Figure 1. Details of the apparatus and the experimental procedure are described elsewhere [18], and a brief explanation is as follows. Liquid CO_2_ filtrated by a filter (2 μm) from the cylinder was supplied by a high-pressure pump (SUPER-201, JASCO, Tokyo, Japan) into the pressure-tight column (8.3 cm^3^) for solubility measurements through a preheated coil placed in an oven. We loaded the solid dye into the column with glass beads and glass wool to fully make contact with scCO_2_. The oven was temperature-controlled within ± 0.1 °C. Pressure of the column was ± 0.1 MPa by a back-pressure regulator. The flow line from the oven to the back pressure regulator was kept hot by a flexible ribbon heater to prevent clogging due to solid dye deposition by depressurization. The dye dissolved into scCO_2_ was collected in a trap filled with ethanol. The concentration of dye in ethanol solution was determined by UV-spectroscopy (BioSpec-1600, Shimadzu, Kyoto, Japan). The volume of CO_2_ gas was measured by a wet gas flow meter (WK-NK-1B, Shinagawa, Tokyo, Japan).

In the experimental procedure, CO_2_ flows in the by-pass line until reaching the scCO_2_ condition. After a steady state, scCO_2_ was turned into the column by a 6-way valve and was provided 20–40 min to make an equilibration. After the measurements, the dye precipitated in the flow line was cleaned to collect in the trap filled with ethanol.

The solubility of dyestuff in supercritical CO_2_ was calculated by Equation (1).
(1)$$y_{2}=\frac{n_{S}}{n_{\mathrm{CO}2}+n_{S}}$$
where *n*_s_ is the number of moles of solute calculated from the UV calibration curve pre-determined by the mass of solute ethanol mixture and *n*_CO2_ is the mole number of CO_2_ calculated by
(2)$$n_{\mathrm{CO}2}=\frac{\left(P_{0}-P_{\mathrm{water}}\right)V_{g}}{RT_{g}}$$

$V_{g}$ is the volume of CO_2_ gas measured at $T_{g}$ by a wet gas meter, *P*_water_ is the vapor pressure of water in a gas meter, and *P*_0_ is the atmospheric pressure. The experimental uncertainties of solubility data were less than ±5.0%.

### 2.3. Dyeing Buttons

The experimental apparatus previously developed for dyeing plastic buttons in supercritical carbon dioxide was modified as shown in Figure 2. Compared to the previous measurements [17], we measured to extend the experimental data necessary for the multiple regression over the whole ranges of pressure, temperature, and treatment time of dyeing for 8 to 14 MPa, 40 to 55 °C, and 30 to 60 min for acrylic, 10 to 16 MPa, 110 to 125 °C, and 20 to 60 min for polyester, 8 to 14 MPa 80 to 110 °C, 20 to 60 min for casein, and 21 to 29 MPa, 110 to 130 °C, and 30 to 100 min for nylon. The experimental ranges for dyeing buttons that do not cause bubbles, cracks, or wrinkles were determined from preliminary measurements. Among the polymers, acrylic is an amorphous and easy-to-color plastic resin, so the dyeing temperature of the acryl button can be lower than the glass transition temperature of acrylic polymer.

The experimental procedure is similar to that previously reported and is briefly described as follows. CO_2_ of a gas cylinder was liquefied by a chiller and fed into a dyeing container using a high-pressure pump (NP-KX-540, Nippon Seimitsu Kagaku, Co. Ltd., Tokyo, Japan). The capacity of the vessel is 50 cm^3^. The dyeing vessel was placed in a constant temperature oven controlled within ±0.1 °C. A hanging shelf was installed in the vessel so that multiple buttons could be dyed at the same time. The pressure was controlled by a back-pressure regulator (26-1700, TESCOM, Atlanta, GA, USA) within ±0.1 MPa. For every measurement, about 0.5 g of dye was loaded in the high-pressure container. A magnetic stirrer bar was used for vigorous mixing in the vessel. After the dyeing, a relief valve was used to reduce atmospheric pressure, preventing bubbles and cracks from appearing in the buttons. As an example, the average rate of the decompression for the acrylic button was 1 MPa/min from dyeing pressure to the critical pressure of CO_2_. Figure 3 depicts the plastic buttons before dyeing. The dimensions of the buttons before and after dyeing in scCO_2_ over the experimental ranges did not noticeably change within +0.45 mm and +0.68 mm in diameter and thickness for acrylic, (−0.1 to +0.18) mm and +0.25 mm in diameter and thickness for polyester, +0.10 mm and (−0.05 to +0.21) mm in diameter and thickness for casein, and +0.12 mm and (−0.07 to +0.06) mm in diameter and thickness for nylon.

Furthermore, we evaluated the dyeing color depth of the buttons by Kubelka–Munk equation [19].
(3)$$K/S={\left(1-R\right)}^{2}/2R$$
where *K* is the absorption coefficient of light, *S* is the scattering coefficient of light, and the surface reflectance *R* is measured by a spectral color difference meter (NF333, Nippon Denshoku Ltd., Tokyo, Japan) with illuminant D_65_ and visual field 10°. *K*/*S* values can indicate the dyeing color depth of plastic buttons, as the larger the *K*/*S* values, the deeper the dyeing color depth of the buttons. The measurements were repeated three times and averaged. The experimental uncertainties of *K*/*S* values were less than ± 1.0% for nylon, ± 2.5% for polyester, and ± 4.0% for acryl and casein.

## 3. Results

Table 1 lists the experimental values of C.I. Disperse Red 22 in scCO_2_ for the temperature ranges of 80 °C to 125 °C and pressures of 15.0 MPa to 25.0 MPa. The experimental results were correlated with the density models proposed by Mendez-Santiago and Teja (MST) [20], Chrastil [21], and Sung and Shim [22].
(4)$$T\;\ln\left(y_{2}P\right)=A+B\rho +CT$$
(5)$$\ln y_{2}=A+B/T+C$$
(6)$$\ln y_{2}=A+B/T+\left(C+D/T\right)$$
where *y*_2_ is the solubility of the dye in the mole fraction, *T* (K) is the temperature, and ρ (mol/m^3^) is the density of CO_2_ estimated by the Span–Wagner equation of state [23] at every given temperature and pressure condition. Table 2 shows the coefficients A, B, and C of Equations (4) to (6) obtained in fitting the model to the experimental solubility data, and the relative absolute arithmetic deviation (AARD) was evaluated by $100\times \left(\frac{y_{2}\left(exp\right)-y_{2}\left(calc\right)}{y_{2}\left(exp\right)}\right)$. The Sung–Shim model with four parameters gave a better representation of solubility than the other models. Figure 4 compares the experimental results with those calculated by the MST model.

We obtained the total number of experimental *K*/*S* values *N* for synthetic resin buttons made of acrylic (*N* = 22), polyester (*N* = 29), casein (*N* = 24), and nylon (*N* = 27), dyed in scCO_2_ at different temperatures *T*, pressures *P*, and treatment times *t*. The experimental *K*/*S* values were found in Appendix A. Figure 5 shows that the color of the plastic buttons dyed in scCO_2,_ as indicated by the *K*/*S* values, changes with respect to the temperature and pressure as well as the treatment time. Generally, the higher the temperature and pressure, as well as the treatment time, the deeper the color of the buttons, and the degree of variation for color changes depends on the material of the buttons. Comparing the degree of the color change in polyester and acrylic buttons, we found that pressure is more effective than the temperature in polyester buttons dyed in scCO_2,_ and temperature is more valid than the pressure in acrylic buttons dyed in scCO_2_. It is known that, at a dyeing temperature more than the glass transition temperature for plastic polymers, scCO_2_ makes plastic polymer swollen and relaxes the polymer network structure. Then, dye molecules can easily diffuse into the polymer network under the controlled pressure. Conversely, at a dyeing temperature lower than the glass transition temperature of plastic polymers, the polymer network and the structure are controlled principally by the glass transition temperature rather than the pressure. Although the glass transition temperatures of the matrix resin constituting the buttons may not agree somewhat with those of the plastic polymers, the amount of dye sorption into the polyester button significantly changed with the pressure at the higher dyeing temperatures than the glass transition temperature of the polyester polymer. The dyeing temperature of the acrylic button was lower than the glass transition temperature, and the dye sorption on the acrylic button varied greatly depending on the dyeing temperature rather than the pressure.

### 3.1. Response Surface Model

The multiple regression analysis [24,25] was applied to examine a quantitative relationship of the *K*/*S* values for each synthetic resin button made of acrylic, polyester, casein, and nylon, with respect to the temperature *T*, pressure *P*, dyeing treatment time *t*, density of CO_2_
*ρ*, and solubility of dyestuff y_2_ as explanatory variables. We correlated the *K*/*S* values for each synthetic resin button made of acrylic, polyester, casein, and nylon using the following several forms of the regression equation.

At first, the *K*/*S* value can be expressed by Equation (7), in terms of the experimental dyeing conditions of the pressure (MPa), temperature (°C), and treatment time (min) for the plastic buttons.
(7)$$K/S=a_{1}+a_{2}P+a_{3}T+a_{4}t+a_{5}PT+a_{6}Pt+a_{7}Tt+a_{8}P^{2}+a_{9}T^{2}+a_{10}t^{2}$$
where $a_{i}$ is the adjustable parameters obtained fitting the equation to the *K*/*S* experimental values.

Secondly, we correlated the *K*/*S* values by Equation (8), using the density of CO_2_, ρ, calculated by the Span–Wagner equation of the state [23] at the given temperature and pressure conditions.
(8)$$K/S=a_{1}+a_{2}\rho +a_{3}T+a_{4}t+a_{5}\rho T+a_{6}\rho t+a_{7}Tt+a_{8}\rho^{2}+a_{9}T^{2}+a_{10}t^{2}$$

Next, we formulated the *K*/*S* values by the concentration $\left(y_{2}\rho \right)$ of the dye dissolved in scCO_2_, expressed by the product of the solubility of dye and CO_2_ density.
(9)$$K/S=a_{1}+a_{2}y_{2}\rho +a_{3}T+a_{4}t+a_{5}y_{2}\rho T+a_{6}y_{2}\rho t+a_{7}Tt+a_{8}{\left(y_{2}\rho \right)}^{2}+a_{9}T^{2}+a_{10}t^{2}$$

Finally, we correlated the *K*/*S* value by the partial pressure $\left(y_{2}P\right)$ of dye in scCO_2_.
(10)$$K/S=a_{1}+a_{2}y_{2}P+a_{3}T+a_{4}t+a_{5}y_{2}PT+a_{6}y_{2}Pt+a_{7}Tt+a_{8}{\left(y_{2}P\right)}^{2}+a_{9}T^{2}+a_{10}t^{2}$$

The results calculated by Equations (7) to (10) are summarized in Table 3, along with the absolute arithmetic mean deviation, AAD, and the absolute arithmetic relative deviation, AARD. The absolute arithmetic mean deviation AAD and relative deviation AARD between the experimental and calculated values were defined by
(11)$$\;\;AAD=\overset{N}{\underset{i=1}{\sum }}\left|{\left(\frac{K}{S}\right)}_{i,exp}-{\left(\frac{K}{S}\right)}_{i,cal}\right|/N\;$$
(12)$$AARD=100\overset{N}{\underset{i=1}{\sum }}\left|\frac{{\left(\frac{K}{S}\right)}_{i,exp}-{\left(\frac{K}{S}\right)}_{i,cal}}{{\left(\frac{K}{S}\right)}_{i,exp}}\right|$$

*N* is the number of experimental data points used in the regression of the *K*/*S* values for synthetic resin buttons made of acrylic (*N* = 22), polyester (*N* = 29), casein (*N* = 24), and nylon (*N* = 27).

Generally, the better the agreement between the experimental and correlated results, the more the number of parameters in the regression equations increased. Furthermore, even if the terms of CO_2_ density, dye concentration, and dye partial pressure were taken into consideration as in Equations (7) to (10), it is not always possible to improve; however, the polyester button did. As a whole, the *K*/*S* values for the acrylic, polyester, casein, and nylon buttons could be represented successfully by Equations (7) to (10). Table 4 lists the equation and the parameters for the buttons obtained by Equation (7) with the least AAD between the experimental and correlated results, as shown in Table 3.

Figure 6 compares the experimental results for each button with those calculated by the multiple regression parameters shown in Table 4. The blue solid lines indicate errors of the deviations between the experimental and calculated ones within +/− 20%. The experimental data of each button were indicated by red solid points. If the solid plots are on the diagonal line, the calculated results correspond to the experimental values.

Figure 7 shows the contour of the *K*/*S* values for the acrylic, polyester, casein, and nylon buttons calculated by Equation (7) as a function of the dyeing treatment time. The *K*/*S* values for all the buttons increased with the treatment time, depending on the characteristic properties of the materials, as the pressure and temperature increased. Especially in the case of acrylic buttons, whose dyeing temperature is lower than the glass transition temperature, the *K*/*S* change at a dyeing time of 50 to 60 min depended largely on the pressure. For PET, casein, and nylon buttons with dyeing temperatures higher than the glass transition temperatures, the *K*/*S* values changed depending on both the temperature and pressure. The *K*/*S* value for PET has a saddle point and changed overall with the dyeing time and rapidly increased over the dyeing time of 40 to 60 min. Casein changed like a trough over time, significantly at the dyeing time of 40 to 60 min. The *K*/*S* value of the nylon button changed overall on the ridge over the dyeing time, although the range of *K*/*S* values is narrow. As mentioned above, it was experimentally clarified that the change in the *K*/*S* value is closely related to the pressure, temperature, CO_2_ density, solubility of the dye in CO_2_, and the dye transfer kinetics into the buttons.

Using the regression equations with the coefficients given in Table 4, we could estimate the optimum operating conditions of temperature, pressure, and dyeing treatment time by maximizing the *K*/*S* values for plastic buttons made of acrylic, polyester, nylon, and casein. The maximum *K*/*S* values were obtained as 47.96 at 55 °C, 14.0 MPa, and 51 min for acrylic, 47.04 at 125 °C, 16.0 MPa, and 60 min for polyester, 44.76 at 110 °C, 14 MPa, and 60 min for casein, and 14.28 at 122 °C, 29.0 MPa, and 75 min for nylon. The results were comparable to the optimum *K*/*S* values determined using the experimental data points (*N* = 9) by the experimental design method reported previously [17], as 46.03 at 50 °C, 14 MPa, and 40 min for acrylic, 40.56 at 120 °C, 12 MPa, 30 min for polyester, 43.25 at 80 °C, 10 MPa, and 30 min for casein, and 15.06 at 125 °C, 29 MPa, and 80 min for nylon.

### 3.2. Dye-Sorption Model with Dye Diffusion into Buttons

We consider a simplified model for the process of dyeing plastic buttons with scCO_2_, where the dye molecule is sufficiently saturated in supercritical carbon dioxide and can reach the button surface without being affected by the boundary film resistance between the supercritical phase and the button. The dye reached the button surface and impregnated into the button instantly by the driving force of the concentration. Since the time required for the dye to diffuse into the button is longer than the time for the adsorption equilibrium on the button surface, the amount of dye retained in the buttons is controlled by the diffusion of the dye in the button. Eventually, we can formulate the *K*/*S* values by combining the dye solubility in scCO_2_, the adsorption equilibrium on the button surface, and the diffusion into buttons with the amount of dye sorption in the button. Following the Fickian diffusion equation [26] for a semi-infinite medium whose surface is maintained at a constant concentration, the mass diffusion of dye into buttons is given by
(13)$$\frac{\partial C}{\partial t}=\frac{\partial }{\partial x}\left(D\frac{\partial C}{\partial x}\right)$$
with initial and boundary conditions of
(14)t=0, x≥0     C=0
(15)$$t\geq 0,\;x=0\;\;\;\;\;C=C_{0}$$
(16)t≥0, x→∞      C=0
where *C* is the dye concentration, $C_{0}$ the initial concentration on the surface of the button, *x* is the diffusion distance from the surface, and *D* is the diffusion constant of dye into the button. The analytical solution of the dye concentration in the button is obtained by
(17)$$\frac{C}{C_{0}}=\mathrm{erfc}\left(\frac{x}{2\sqrt{Dt}}\right)$$

The amount of dye sorption in the button $M_{t}$ can be calculated by the integration of the dye flux through the diffusion time.
(18)$$M_{t}=\int_{0}^{t}{\left(D\frac{\partial C}{\partial x}\right)}_{x=0}dt=2C_{0}\sqrt{\frac{Dt}{\pi }}$$

We assume that the initial dye concentration is given by the dye adsorption on the surface of the button, as an example is expressed by the Langmuir adsorption isotherm [27] in terms of the dye molar concentration $\left(y_{2}\rho \right)$ in scCO_2_,
(19)$$C_{0}=\frac{n_{\infty }k\left(y_{2}\rho \right)}{1+k\left(y_{2}\rho \right)}$$
where *k* and $n_{\infty }$ are Langmuir constants. Finally, we can formulate the K/S values by combing Equations (18) and (19), since the K/S value is proportional to the amount of dye sorption in the button.
(20)$$\frac{K}{S}=\alpha M_{t}=2\alpha \left\{\frac{n_{\infty }k\left(y_{2}\rho \right)}{1+k\left(y_{2}\rho \right)}\right\}\sqrt{\frac{Dt}{\pi }}=\left\{\frac{a_{1}a_{2}\left(y_{2}\rho \right)}{1+a_{2}\left(y_{2}\rho \right)}\right\}\sqrt{t}$$
where α is the proportional constant of *K*/*S* and $M_{t}$ and $a_{i}$ the adjustable parameters. Table 5 shows the adsorption isotherms [28,29,30,31] used in the present work. The Henry-type, Langmuir-type, Freundlich-type, and Radke–Prausnitz, and their adsorption isotherms with temperature-dependent coefficients in the formula given by Equations (21) to (36), as well as the BET equation for multi-layer adsorption and the Dubinin–Astakhov (DA) equation were applied.

**Table 5 materials-16-00907-t005:** Absolute arithmetic mean deviation and relative mean deviation for K/S between experimental and calculated values obtained by Equations (20) to (36).

K/S	Equation	Absolute Arithmetic Mean Deviation, AAD [-]				Absolute Relative Deviation, AARD [%]			
		Acryl	PET	Casein	Nylon	Acryl	PET	Casein	Nylon
$\left\{a_{1}+a_{2}/T+a_{3}/T^{2}\right\}\left(y_{2}\rho \right)\sqrt{t}$	(21)	18.5	10.2	22.5	3.9	67.7	59.2	69.6	29.4
$\{a_{1}\left(y_{2}\rho \right)+a_{2}{\left(y_{2}\rho \right)}^{2}+a_{3}{\left(y_{2}\rho \right)}^{3}\}\sqrt{t}$	(22)	13.2	12.8	14.9	2.4	48.5	39.5	46.6	18.5
$\left\{\frac{a_{1}a_{2}\left(y_{2}\rho \right)}{1+a_{2}\left(y_{2}\rho \right)}\right\}\sqrt{t}$	(20)	6.7	5.6	6.1	2.2	22.1	15.8	18.2	16.6
$\left\{\frac{a_{1}a_{2}\left(y_{2}\rho \right)}{1+a_{2}{\left(y_{2}\rho \right)}^{a_{3}}}\right\}\sqrt{t}$	(23)	5.8	5.6	5.4	2.2	19.0	15.9	15.9	16.4
$\left\{a_{1}\left(y_{2}\rho \right)+\frac{a_{2}a_{3}\left(y_{2}\rho \right)}{1+a_{3}\left(y_{2}\rho \right)}\right\}\sqrt{t}$	(24)	5.8	5.6	5.3	2.2	19.1	15.8	15.8	16.4
$\left\{\frac{(a_{1}+a_{2}/T)a_{3}\left(y_{2}\rho \right)}{1+a_{3}\left(y_{2}\rho \right)}\right\}\sqrt{t}$	(25)	6.7	5.6	5.5	2.2	22.1	15.8	16.3	16.4
$\left\{\frac{a_{1}a_{2}e^{-a_{3}/T}\left(y_{2}\rho \right)}{1+a_{2}e^{-a_{3}/T}\left(y_{2}\rho \right)}\right\}\sqrt{t}$	(26)	6.7	5.6	5.5	2.2	22.1	15.8	16.3	16.4
$\left\{\frac{(a_{1}+a_{2}/T)a_{3}e^{-a_{4}/T}\left(y_{2}\rho \right)}{1+(a_{1}+a_{2}/T)a_{3}e^{-a_{4}/T}\left(y_{2}\rho \right)}\right\}\sqrt{t}$	(27)	6.3	5.6	5.5	2.2	20.9	15.8	16.2	16.4
${\left\{\frac{a_{2}{\left(a_{3}y_{2}\rho \right)}^{a_{1}}}{1+{\left(a_{3}y_{2}\rho \right)}^{a_{1}}}\right\}}^{a_{4}/a_{1}}\sqrt{t}$	(28)	6.1	5.6	5.5	2.2	20.1	15.8	16.2	16.4
$a_{1}{\left(y_{2}\rho \right)}^{a_{2}}\sqrt{t}$	(29)	5.8	5.6	5.4	2.2	19.0	15.9	15.9	16.4
$(a_{1}+a_{2}/T){\left(y_{2}\rho \right)}^{a_{3}}\sqrt{t}$	(30)	5.8	5.6	5.4	2.2	19.0	15.9	15.9	16.4
$a_{1}{\left(y_{2}\rho \right)}^{\left(a_{2}+a_{3}/T\right)}\sqrt{t}$	(31)	5.8	5.6	5.4	2.2	19.0	15.9	15.9	16.4
$(a_{1}+a_{2}/T){\left(y_{2}\rho \right)}^{\left(a_{3}+a_{4}/T\right)}\sqrt{t}$	(32)	5.7	5.6	5.4	2.2	18.9	16.0	15.9	16.1
$\left\{\frac{a_{1}a_{2}\left(y_{2}\rho \right)}{a_{3}-\left(y_{2}\rho \right)}\frac{1}{(a_{2}-1)\left(y_{2}\rho \right)+a_{3}}\right\}\sqrt{t}$	(33)	5.9	5.6	5.3	2.2	19.5	15.8	15.8	16.5
$\left\{\frac{a_{1}a_{2}e^{-a_{4}}\left(y_{2}\rho \right)}{a_{3}-\left(y_{2}\rho \right)}\frac{1}{(a_{2}e^{-a_{4}}-1)\left(y_{2}\rho \right)+a_{3}}\right\}\sqrt{t}$	(34)	5.9	5.6	5.3	2.2	19.5	15.8	15.8	16.6
$\left\{\frac{a_{1}a_{2}x}{1-x}\cdot \frac{1-\left(a_{4}+1\right)x^{a_{4}}+a_{4}x^{a_{4}+1}}{1+\left(a_{2}-1\right)x^{a_{4}}-a_{2}x^{a_{4}+1}}\right\}\sqrt{t}$ $x=\left(y_{2}\rho \right)/a_{3}$	(35)	6.2	5.6	5.3	2.2	20.4	15.8	15.8	16.4
$a_{1}\exp\left[-{\left\{\frac{RT}{a_{2}}\ln\left(\frac{a_{3}}{y_{2}\rho }\right)\right\}}^{a_{4}}\right]\;\sqrt{t}$	(36)	5.8	5.8	5.5	2.2	19.1	16.4	16.4	16.4

Table 5 summarizes the absolute arithmetic mean deviation, AAD and relative mean deviation, AARD for K/S between the experimental and calculated results. A good agreement between the experimental and calculated ones was obtained from the sorption models with Langmuir, Freundlich, Radke–Prausnitz, and Henry–Langmuir adsorption isotherms. On behalf of the good correlated results, the parameters $a_{i}$ in the model obtained fitting Equation (23) to the experimental data were listed in Table 6. It found that the results could not always be noticeably improved even when increasing the number of temperature-dependent parameters used in the isotherms. In addition, the multi-layer models of the BET equation and the DA equation yielded comparable results to the single-layer adsorption models.

Furthermore, we assume the dye adsorption on the surface of the button expressed by Langmuir adsorption isotherm in terms of the dye partial pressure $\left(y_{2}P\right)$ in scCO_2_ in place of the dye molar concentration $\left(y_{2}\rho \right)$ of Equation (17).


(37)
$$C_{0}=\frac{n_{\infty }k\left(y_{2}P\right)}{1+k\left(y_{2}P\right)}$$


Combining Equations (18) and (37), we obtain the *K*/*S* value as


(38)
$$K/S=2\alpha \left\{\frac{n_{\infty }k\left(y_{2}P\right)}{1+k\left(y_{2}P\right)}\right\}\sqrt{\frac{Dt}{\pi }}=\left\{\frac{a_{1}a_{2}\left(y_{2}P\right)}{1+a_{2}\left(y_{2}P\right)}\right\}\sqrt{t}$$


The sorption models with $\left(y_{2}P\right)$ expressed by Equations (39) to (53) were shown in Table 7, along with the deviations between the experimental and calculated results. As a representative one, Table 8 lists the parameters $a_{i}$ obtained in fitting Equation (47) to the experimental data. The results obtained by the sorption model with the adsorption isotherm in terms of the dye partial pressure in scCO_2_ were agreed to be comparable with those calculated by the model with the dye molar concentration in scCO_2_. Furthermore, the amount of sorption could be expressed using the adsorption formula with a small number of parameters, compared with the results calculated using the adsorption amount expressed by the polynomial of the temperature, pressure, CO_2_ density, and dye solubility, as shown in Appendix B. These results suggest that the dye-sorption model has good potential in the coloration processing of plastic buttons with scCO_2_.

## 4. Conclusions

We carried out the dyeing of plastic buttons made of acrylic, polyester, casein, and nylon in supercritical carbon dioxide at temperature ranges of 40 to 55 °C, pressures of 8 to 14 MPa, and dyeing times of 30 to 60 min for acrylic (*N* = 22); 110 to 125 ℃, 10 to 16 MPa, and 20 to 60 min for polyester (*N* = 29); 80 to 110 °C, 8 to 14 MPa, and 30 to 60 min for casein (*N* = 24); and 110 to 130 °C, 21 to 29 MPa, and 30 to 100 min for nylon (*N* = 29). We examined the dyeing color depth of the buttons using *K*/*S* values for the dyeing of the buttons evaluated using the Kubelka–Munk equation. Moreover, we measured the solubility of C.I. Disperse Red 22 in scCO_2_ over the temperature ranges of 80 °C to 125 °C and pressures of 15.0 MPa to 25.0 MPa. The experimental data were correlated within the experimental uncertainties using the density models previously proposed. The *K*/*S* values for the plastic buttons made of acrylic, polyester, casein, and nylon dyed in scCO_2_ were represented well by the polynomials in the temperature, pressure, density of scCO_2_, dyeing treatment time, and with the solubility of the disperse dyestuff. It found that the *K*/*S* values for the buttons dyeing in scCO_2_ could correlate accurately with the polynomials and the optimum *K*/*S* values agreed, comparably with those determined by the experimental design method with the experimental data points (*N* = 9).

From the viewpoint of mass transfer for the dye transport into the polymer, we formulated the amount of dye sorption in the button with a dye-sorption model by combining the dye solubility in scCO_2_, adsorption equilibrium on the button surface, and Fickian diffusion into the buttons. We found that the sorption model worked reasonably within the framework of Fickian diffusion in the representation of the dyeing buttons in scCO_2_. The *K*/*S* results correlated fairly well with a minimum number of parameters, if the adsorption isotherms used in the sorption model could be expressed in terms of both the concentration and partial pressure of the dye that act as the driving force of the phenomena in scCO_2_. The single-layer adsorption models of Langmuir, Freundlich, Radke–Prausnitz, and Henry–Langmuir could reproduce the experimental results successfully, even in comparison with the multi-layer models of the BET and Dubinin-Astakhov. Finally, we could represent the *K*/*S* results by the dye-sorption model, suggesting the good potential of the dye-sorption model to be developed in the coloring processing of plastic buttons with scCO_2_.

## Figures and Tables

**Figure 1 materials-16-00907-f001:**
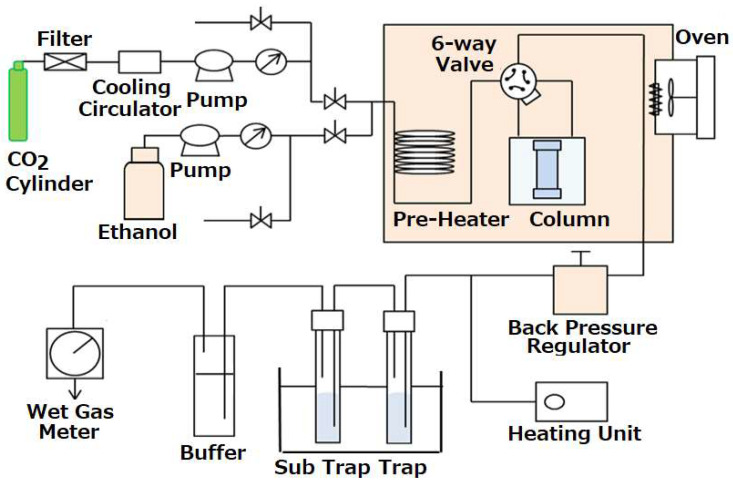
Experimental apparatus for solubility measurement.

**Figure 2 materials-16-00907-f002:**
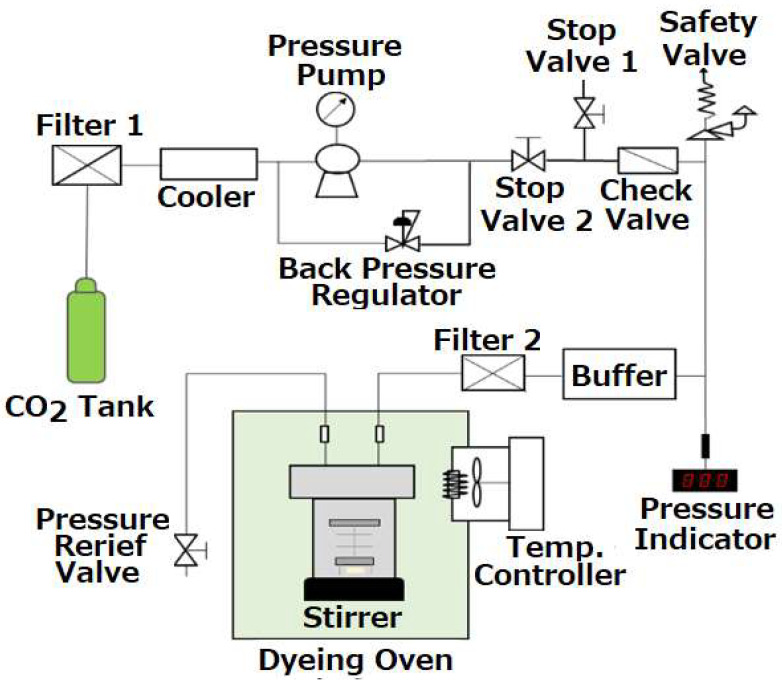
Supercritical carbon dioxide dyeing apparatus.

**Figure 3 materials-16-00907-f003:**
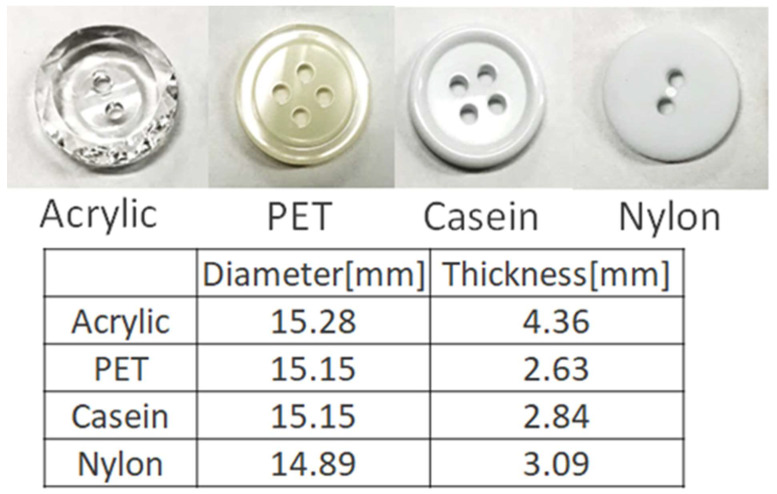
Plastic buttons and the dimension.

**Figure 4 materials-16-00907-f004:**
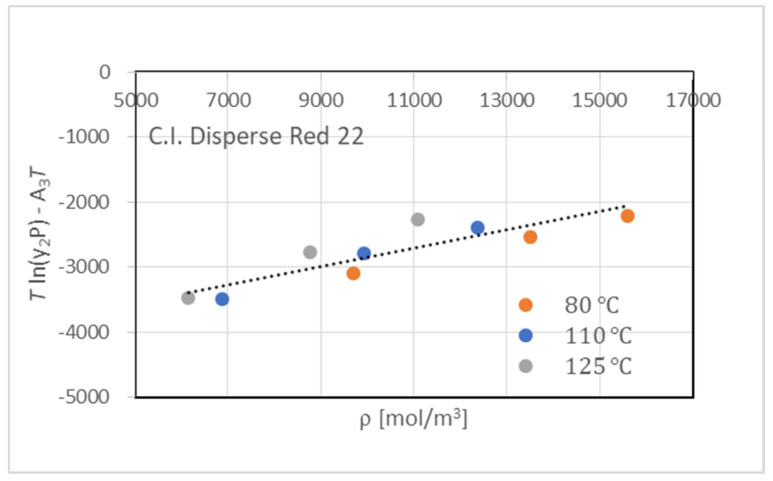
Solubility of C.I. Disperse Red 22 in scCO_2_ correlated by MST model.

**Figure 5 materials-16-00907-f005:**
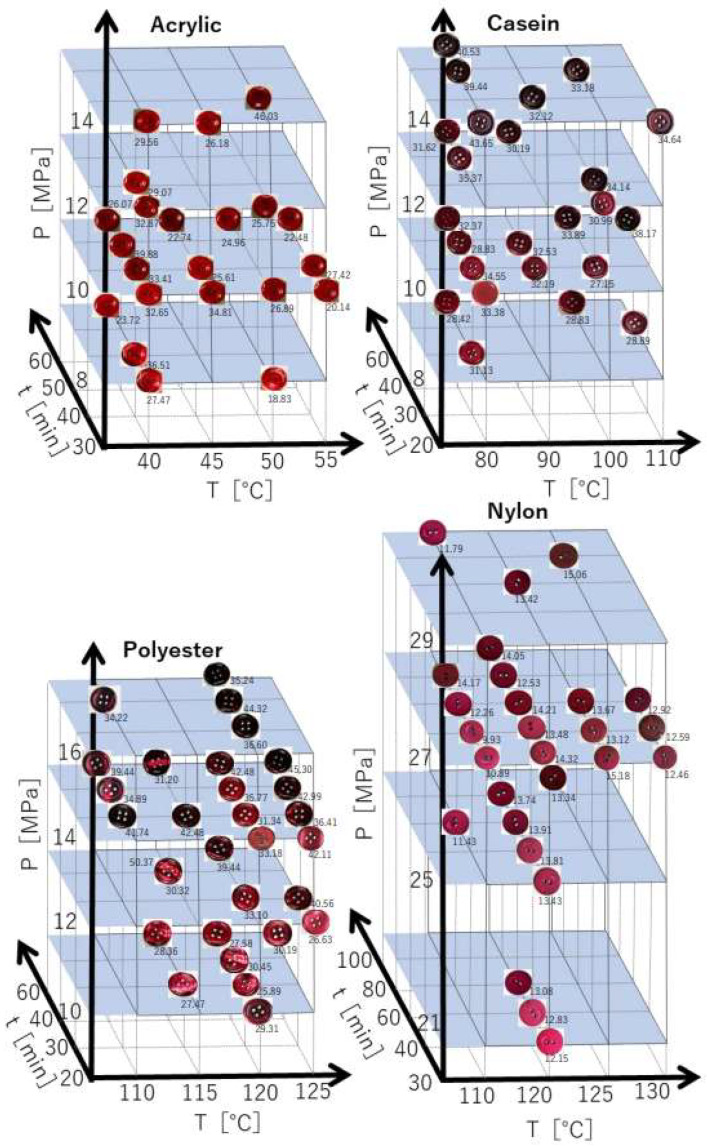
Color change (*K*/*S*) of acrylic, polyester, nylon, and casein buttons after scCO_2_ dyeing.

**Figure 6 materials-16-00907-f006:**
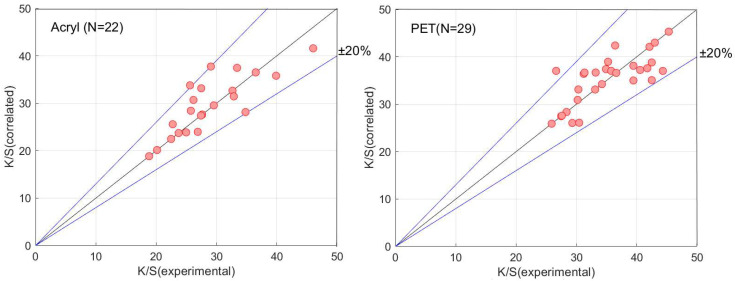
Regression results of acrylic, polyester, casein, and nylon buttons.

**Figure 7 materials-16-00907-f007:**
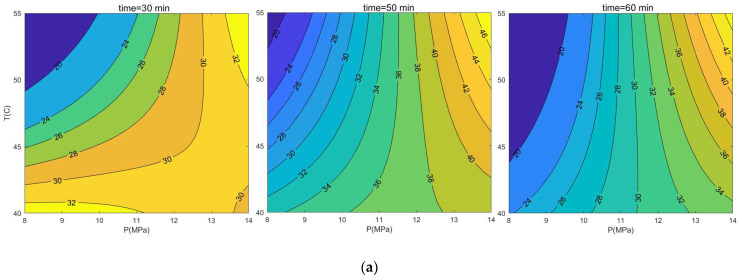
Contour of *K*/*S* values for acrylic (**a**), polyester (**b**), casein (**c**), and nylon (**d**) buttons as a function of dyeing time.

**Table 1 materials-16-00907-t001:** Solubility of C.I. Disperse Red 22 in scCO_2_.

*T* [°C]	*P* [MPa]	ρ [mol/m^3^]	*y*_2_ (×10^5^)
80	15	9705.89	1.05
80	20	13494.5	3.76
80	25	15592.5	7.70
110	15	6887.06	0.730
110	20	9926.13	3.40
110	25	12376.3	7.73
125	15	6151.74	1.06
125	20	8766.43	4.74
125	25	11084.2	13.6

**Table 2 materials-16-00907-t002:** Coefficients of the model parameters and absolute arithmetic relative deviations of the solubility data.

	*A*	*B*	*C*	*D*	AARD (%)
Chrastil Model	−33.091	−5786.58	4.1310		13.2
MST Model	−9997.8	0.19696	27.3672		20.4
Sung–Shim Model	3.6208	−19785.5	0.1992	1498.3	11.8

**Table 3 materials-16-00907-t003:** Absolute arithmetic mean deviation and relative mean deviation for *K*/*S* by Equations (7) to (10).

	Absolute Arithmetic Mean Deviation, AAD [-]				Absolute Relative Deviation, AARD [%]			
	Acryl	PET	Casein	Nylon	Acryl	PET	Casein	Nylon
Equation (7)	2.60	2.78	1.76	0.46	8.8	8.5	4.4	3.5
Equation (8)	2.66	2.79	1.76	0.46	8.1	8.2	4.4	3.5
Equation (9)	2.61	2.84	1.86	0.46	9.2	8.1	4.8	3.5
Equation (10)	2.71	3.84	2.71	0.46	9.6	8.1	4.9	3.5

**Table 4 materials-16-00907-t004:** Regression parameters of Equation (7).

	a_1_ a_6_	a_2_ a_7_	a_3_ a_8_	a_4_ a_9_	a_5_ a_10_
Acrylic	177.707	−12.2007	−4.79155	1.36692	0.282419
	0.073609	0.016355	−0.086532	6.76624 × 10^−3^	−0.032495
PET	1318.078	−5.56893	−21.3484	−0.780510	0.158916
	3.68418 × 10^−3^	5.32500 × 10^−3^	−0.454877	0.082058	1.78842 × 10^−3^
Casein	249.021	1.63815	−4.24356	−1.77318	−0.014261
	0.014635	0.011041	0.029126	0.020642	−7.58398 × 10^−3^
Nylon	−260.714	−0.196155	4.48523	0.135269	1.01555 × 10^−3^
	2.34139 × 10^−4^	−8.5074 × 10^−4^	−1.8022 × 10^−3^	−0.019397	−2.560757 × 10^−4^

Dimensions of *P*, *T*, ρ, and *t* used in the equation were expressed by MPa, °C, mol/m^3^, and min.

**Table 6 materials-16-00907-t006:** Parameters of regression equation by Equation (23).

	a_1_	a_2_	a_3_
Acrylic	5.29100	4701150	0.93684
PET	7.99087	675032	0.89351
Casein	7.37227	817877	0.92644
Nylon	1.76363	560.865	1.11226

Dimensions of *T*, ρ, and *t* used in the equation were expressed by K, mol/m^3^, and min.

**Table 7 materials-16-00907-t007:** Absolute arithmetic mean deviation and relative mean deviation for K/S between experimental and calculated values obtained by Equations (38) to (53).

K/S	Equation	Absolute Arithmetic Mean Deviation, AAD [-]				Absolute Relative Deviation, AARD [%]			
		Acryl	PET	Casein	Nylon	Acryl	PET	Casein	Nylon
$\left\{a_{1}+a_{2}/T+a_{3}/T^{2}\right\}\left(y_{2}P\right)\sqrt{t}$	(39)	18.5	19.2	21.9	3.8	67.7	56.8	67.7	28.6
$\{a_{1}\left(y_{2}P\right)+a_{2}{\left(y_{2}P\right)}^{2}+a_{3}{\left(y_{2}P\right)}^{3}\}\sqrt{t}$	(40)	18.5	19.2	21.9	2.9	67.7	56.8	67.7	22.0
$\left\{\frac{a_{1}a_{2}\left(y_{2}P\right)}{1+a_{2}\left(y_{2}P\right)}\right\}\sqrt{t}$	(38)	6.7	5.5	5.5	2.2	22.1	15.7	16.4	16.4
$\left\{\frac{a_{1}a_{2}\left(y_{2}P\right)}{1+a_{2}{\left(y_{2}P\right)}^{a_{3}}}\right\}\sqrt{t}$	(41)	5.8	5.6	5.3	2.8	19.0	15.8	15.8	21.2
$\left\{a_{1}\left(y_{2}P\right)+\frac{a_{2}a_{3}\left(y_{2}P\right)}{1+a_{3}\left(y_{2}P\right)}\right\}\sqrt{t}$	(42)	5.8	5.5	5.4	2.8	19.1	15.7	16.0	21.1
$\left\{\frac{(a_{1}+a_{2}/T)a_{3}\left(y_{2}P\right)}{1+a_{3}\left(y_{2}P\right)}\right\}\sqrt{t}$	(43)	6.6	5.5	5.5	2.0	22.1	15.7	16.3	15.2
$\left\{\frac{a_{1}a_{2}e^{-a_{3}/T}\left(y_{2}P\right)}{1+a_{2}e^{-a_{3}/T}\left(y_{2}P\right)}\right\}\sqrt{t}$	(44)	6.7	5.5	5.5	2.2	22.1	15.7	16.4	16.4
$\left\{\frac{(a_{1}+a_{2}/T)a_{3}e^{-a_{4}/T}\left(y_{2}P\right)}{1+(a_{1}+a_{2}/T)a_{3}e^{-a_{4}/T}\left(y_{2}P\right)}\right\}\sqrt{t}$	(45)	6.6	5.5	5.5	2.0	21.6	15.7	16.3	15.1
${\left\{\frac{a_{2}{\left(a_{3}y_{2}P\right)}^{a_{1}}}{1+{\left(a_{3}y_{2}P\right)}^{a_{1}}}\right\}}^{a_{4}/a_{1}}\sqrt{t}$	(46)	5.8	5.5	5.3	2.2	19.1	15.7	15.8	16.4
$a_{1}{\left(y_{2}P\right)}^{a_{2}}\sqrt{t}$	(47)	5.8	5.5	5.3	2.2	19.0	15.7	15.8	16.4
$(a_{1}+a_{2}/T){\left(y_{2}P\right)}^{a_{3}}\sqrt{t}$	(48)	5.8	5.5	5.3	2.2	19.0	15.7	15.8	16.8
$a_{1}{\left(y_{2}P\right)}^{\left(a_{2}+a_{3}/T\right)}\sqrt{t}$	(49)	5.8	5.5	5.3	2.2	19.0	15.7	15.8	16.6
$(a_{1}+a_{2}/T){\left(y_{2}P\right)}^{\left(a_{3}+a_{4}/T\right)}\sqrt{t}$	(50)	5.8	5.5	5.3	2.2	19.0	15.7	15.8	16.6
$\left\{\frac{a_{1}a_{2}\left(y_{2}P\right)}{a_{3}-\left(y_{2}P\right)}\frac{1}{(a_{2}-1)\left(y_{2}P\right)+a_{3}}\right\}\sqrt{t}$	(51)	5.9	5.5	5.3	3.1	19.5	15.7	15.8	23.6
$\left\{\frac{a_{1}a_{2}e^{-a_{4}}\left(y_{2}P\right)}{a_{3}-\left(y_{2}P\right)}\frac{1}{(a_{2}e^{-a_{4}}-1)\left(y_{2}P\right)+a_{3}}\right\}\sqrt{t}$	(52)	6.7	5.5	5.4	2.2	22.2	15.7	16.1	16.7
$\left\{\frac{a_{1}a_{2}x}{1-x}\cdot \frac{1-\left(a_{4}+1\right)x^{a_{4}}+a_{4}x^{a_{4}+1}}{1+\left(a_{2}-1\right)x^{a_{4}}-a_{2}x^{a_{4}+1}}\right\}\sqrt{t}$ $x=\left(y_{2}P\right)/a_{3}$	(53)	5.9	5.5	5.3	2.1	19.5	15.7	15.9	16.2

**Table 8 materials-16-00907-t008:** Parameters of regression equation by Equation (47).

	a_1_	a_2_
Acrylic	8.91565	0.0710291
PET	16.2783	0.113623
Casein	12.4299	0.0794947
Nylon	1.70045	0.366208 × 10^−8^

Dimensions of *P*, *T*, ρ, and *t* used in the equation were expressed by MPa, K, mol/m^3^, and min.

## Data Availability

All data in this paper are presented in the form of figures, tables and Appendix A. Anyone who is interested in the details can contact the corresponding author (tamura@se.kanazawa-u.ac.jp) directly.

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
