# Peer review of "Coloration Modeling and Processing of Commodity Plastic Buttons in Supercritical Carbon Dioxide"

_materials, 2023, doi:10.3390/ma16030907_

Round 1

Reviewer 1 Report

Reviewer comments and suggestions:

 The manuscript titled “Coloration Modeling and Processing of Commodity Plastic Buttons in Supercritical Carbon Dioxide” by Kota Kobayashi, Tierong Bai, Kazuhiro Tamura, K. Tada, Jun Yan and Laijiu Zheng. Submitted to Materials

 The authors’ work aim to study the color processing of the plastic buttons of acrylic, polyester, and casein in scCO₂ to control the degree of dyeing for the plastic buttons.

 The work suggested experimentally that the solubility of dyestuff increases with increasing the density of the super critical CO2 as green solvent.

 This work evaluated the dyeing color depth by Kubelka-Munk equation.

 1. The reference 19 and 20 are interchanged, please correct them.

 2. Authors may give explanation for selecting the range of temperature and pressure for dyeing the different buttons type. It seems there is no correlation with glass transition temperatures.

 3. Why the dyeing temperature is lower than the Tg for acrylic button only?, provide suitable reason.

 4. Page6, line 177 and 178: the dyeing temperature over the ranges of 100 to 130 for nylon was 177 lower than the glass transition temperature Tg = 64.4 of nylon. Correct the statement.

 5. Use conventional scCO2 in the place of SC-CO2 for representing supercritical CO2.

 6. For a given condition, at particular temperature, pressure, and time, comparison may be discussed for different buttons studied in this work.

 The manuscript is well written comparing the experimental results with models. The work is more relevant for the environmentally greener technology development. The manuscript may be considered for the publication after addressing the suggestions given in the reviewer’s comments.

 Good Luck and Best Wishes

Author Response

Thank you very much for the valuable comments on the revisions. We revised the manuscript to meet the Reviewers’ comments’ as follows.

Reviewer 1

  1. The reference 19 and 20 are interchanged, please correct them.

Thank you for the comment. We cited references 19 and 20 correctly.

  1. Authors may give explanation for selecting the range of temperature and pressure for dyeing the different buttons type. It seems there is no correlation with glass transition temperatures.

Thank you for the comment. We added the relevant sentences.

The experimental ranges for dyeing buttons that do not cause bubbles, cracks, and wrinkles were determined from preliminary measurements.

  1. Why the dyeing temperature is lower than the Tgfor acrylic button only?, provide suitable reason.

We revised the statement and inserted the relevant explanation in the text.

Among the polymers, acrylic is an amorphous and easy-to-to-color plastic resin, so the dyeing temperature of the acryl button can be lower than the glass transition temperature of acrylic polymer.

  1. Page6, line 177 and 178: the dyeing temperature over the ranges of 100 to 130 ℃for nylon was 177 lower than the glass transition temperature Tg = 64.4 ℃ of nylon. Correct the statement.

We revised the statement correctly and inserted the relevant explanation in the text.

Although the glass transition temperatures of the matrix resin constituting the buttons may not agree somewhat with those of the plastic polymers, the amount of dye sorption into the polyester button significantly changed with pressure at the higher dyeing temperatures than the glass transition temperature of the polyester polymer. The dyeing temperature of acrylic was lower than the glass transition temperature, and the dye sorption on the acrylic button varied greatly depending on the dyeing temperature rather than pressure.

  1. Use conventional scCO2in the place of SC-CO2 for representing supercritical CO2.

We used scCO2 in place of SC-CO2 for representing supercritical CO2.

  1. For a given condition, at particular temperature, pressure, and time, comparison may be discussed for different buttons studied in this work.

 We already mentioned the maximum K/S values of the dyeing buttons before the section 3.2. Dye-Sorption model with dye diffusion into buttons on page 11.

The maximum K/S values were obtained as 47.96 at 55 ℃, 14.0 MPa, and 51 min for acrylic, 47.04 at 125 ℃, 16.0 MPa, and 60 min for polyester, 44.76 at 110℃, 14 MPa, and 60 min for casein, and 14.28 at 122 ℃, 29.0 MPa, and 75 min for nylon.

Reviewer 2 Report

The manuscript deals with the dye process in sc CO2: solubility of the dye is measured. Buttons manufactured using different plastics were dyed in sc CO2 with varying the pressure, temperature and time of the process. Modeling was performed for both solubility results and for the dyeing efficiency. The obtained results are of importance for potential implementation of the sustainable sc CO2 dyeing technology that does not lead to water pollution and can save energy. 

A few issues need to be addressed before publishing. 

  1. Throughout the manuscript the authors refer to glass transition temperatures and other parameters of plastic buttons. For example:

Lines 75-76: The average-molecular mass and 75 polydispersity index and glass transition temperature of the buttons was reported earlier

Line 168: “…the glass transition 168 temperature of plastic buttons…”,

etc.

Yet, glass transition temperature and average-molecular mass are applicable only to polymers themselves, not to the products, such as buttons. Please, rephrase. 

  1. Solubility results and modeling should be presented in the Results section, no the Experimental section. 

  2. Line 141: a relief valve was used to reduce gradually slow the pressure atmospheric 141 pressure preventing bubbles and cracks from appearing in the buttons.

The sentence is hard to follow and is generally confusing - please, rephrese. Further, please elaborate on the pressure release rate - what constitutes a “slow” and “gradual” decompression, is it 1 atm per 1 minute, 1 atm per 10 minutes…?

  1. Lines 142-144: The mass and dimensions of synthetic resin buttons dyed before and after were measured for acrylic, pol-143 yester, casein, and nylon buttons.

I could not find the data in the manuscript. Please, reference the table where the data is presented and briefly discuss it. The important question is: is there any permanent change in dimensions due to swelling in sc CO2? This would be a key issue for potential industrial application of the process. 

  1. Figure 4. 

As presented in the manuscript right now, Figure 4 is of poor resolution. Please discuss the numbers  associated with each button. Please, provide photos of buttons of each type before dyeing - if not in Figure 4, then in supplementary. 

Author Response

Thank you very much for the valuable comments on the revisions. We revised the manuscript to meet the Reviewers’ comments’ as follows.

Reviewer 2 

  1. Throughout the manuscript the authors refer to glass transition temperatures and other parameters of plastic buttons. For example:

Lines 75-76: The average-molecular mass and 75 polydispersity index and glass transition temperature of the buttons was reported earlier

Line 168: “…the glass transition 168 temperature of plastic buttons…”,

etc. Yet, glass transition temperature and average-molecular mass are applicable only to polymers themselves, not to the products, such as buttons. Please, rephrase. 

       We added relevant explanations of glass transition temperatures of polymer and plastic buttons.

       On page 2,

The glass transition temperatures of acrylic, polyester, casein, and nylon polymers were estimated to be 96.2 ℃-101.4 ℃, 53.0 ℃-56.9 ℃, 60.5 ℃, and 64.4 ℃, respectively.

On page 4,

Although the glass transition temperatures of the matrix resin constituting the buttons may not agree somewhat with those of the plastic polymers, the amount of dye sorption into the polyester button significantly changed with pressure at the higher dyeing temperatures than the glass transition temperature of the polyester polymer. The dyeing temperature of acrylic was lower than the glass transition temperature, and the dye sorption on the acrylic button varied greatly depending on the dyeing temperature rather than pressure.

  1. Solubility results and modeling should be presented in the Results section, no the Experimental section. 

We merged the solubility results and modeling into the Results section. Renumbering the Equations and Figure.

  1. Line 141: a relief valve was used to reduce gradually slow the pressure atmospheric 141 pressure preventing bubbles and cracks from appearing in the buttons.

The sentence is hard to follow and is generally confusing - please, rephrese. Further, please elaborate on the pressure release rate - what constitutes a “slow” and “gradual” decompression, is it 1 atm per 1 minute, 1 atm per 10 minutes…?

 We inserted the relevant explanations of the decomposition.

After the dyeing, a relief valve was used to reduce atmospheric pressure preventing bubbles and cracks from appearing in the buttons. As an example, the average rate of the decompression for the acrylic button was 1 MPa/min from dyeing pressure to the critical pressure of CO2.

  1. Lines 142-144: The mass and dimensions of synthetic resin buttons dyed before and after were measured for acrylic, pol-143 yester, casein, and nylon buttons. I could not find the data in the manuscript. Please, reference the table where the data is presented and briefly discuss it. The important question is: is there any permanent change in dimensions due to swelling in sc CO2? This would be a key issue for potential industrial application of the process. 

We added the detailed explanations of the changes for the dimension of the buttons before and after dyeing the buttons.

The dimensions of the buttons before and after dyeing in scCO2 over the experimental ranges did not noticeably change within +0.45 mm and +0.68 mm in diameter and thickness for acrylic, (- 0.1 to +0.18) mm and +0.25 mm in diameter and thickness for polyester, +0.10 mm and (-0.05 to +0.21) mm in diameter and thickness for casein, and +0.12 mm and (-0.07 to +0.06) mm in diameter and thickness for nylon.

  1. Figure 4. 

As presented in the manuscript right now, Figure 4 is of poor resolution. Please discuss the numbers associated with each button. Please, provide photos of buttons of each type before dyeing - if not in Figure 4, then in supplementary. 

 We inserted a new Figure 4 with a high resolution and the relevant explanations of the number and values of K/S in Figure 4. The photos before dyeing and the dimension of the buttons was shown in new Figure 3.